# Cryptic Divergence of *Rochia nilotica* (Gastropoda: Tegulidae) from Chuuk Lagoon, Federated States of Micronesia, Revealed by Morphological and Mitochondrial Genome Analyses

**DOI:** 10.3390/ani15233471

**Published:** 2025-12-02

**Authors:** Jong-Seop Shin, Yeong-Ji Park, Changju Lee, Heung-Sik Park, Dongsung Kim, Chi-une Song, Kyungman Kwon, Sang-Woo Hur, Byung-Hwa Min, June Kim, Hyun-Sung Yang

**Affiliations:** 1Tropical & Subtropical Research Center, Korea Institute of Ocean Science & Technology (KIOST), Jeju 63349, Republic of Korea; whdtjqtls@kiost.ac.kr (J.-S.S.);; 2KIOST School, University of Science and Technology (UST), Daejeon 34113, Republic of Korea; 3Korea South Pacific Ocean Research Center, Korea Institute of Ocean Science & Technology (KIOST), Busan 49111, Republic of Korea; 4Jeju Research Institute, Korea Institute of Ocean Science & Technology (KIOST), Jeju 63349, Republic of Korea; 5Department of Life Science, Chung-Ang University, Seoul 06974, Republic of Korea; 6Aquafeed Research Center, National Institute of Fisheries Science, Pohang 37517, Republic of Korea; 7Research Cooperation Division, National Institute of Fisheries Science, Busan 46083, Republic of Korea

**Keywords:** Chuuk Atoll, cryptic species, Trochoidea, phylogenetics, marine resource

## Abstract

*Rochia nilotica* is a tropical marine gastropod inhabiting Indo-Pacific coral reefs and is an important resource for many Pacific Island nations, where it is used both as food and as a source of shell material for traditional crafts. To better understand the genetic identity of this species in remote regions, we collected individuals from Weno Island in Chuuk Atoll, Federation of Micronesia, and analyzed their complete mitochondrial genome. We also compared two commonly used DNA markers (COX1 and 16S rRNA) together with 13 protein-coding genes from the mitochondrial genome to reconstruct their evolutionary relationships. The genetic results showed that the Chuuk population forms congruent topology with *R. nilotica* from other regions, but it also occupies a distinct position within the *Rochia* group. This pattern suggests that long-term geographic isolation of Chuuk Atoll may have shaped unique genetic characteristics in this population. These findings highlight the importance of combining DNA data with traditional shell-based identification to correctly recognize species. Such integrative approaches are essential for conserving marine biodiversity and supporting sustainable aquaculture and fisheries management in isolated reef ecosystems.

## 1. Introduction

*Rochia nilotica* (Linnaeus, 1767), formerly known as *Trochus niloticus* Linnaeus, 1767, commonly known as the top shell, is a large marine gastropod belonging to the family Tegulidae, typically inhabiting shallow coral reef environments. This species has a broad distribution across the Indo-Pacific region, with its type locality in the Indian Ocean, and occurs from East Africa to northern Australia, Southeast Asia, and numerous South Pacific islands [1,2,3]. Adult individuals can attain shell heights up to 120 mm, and are primarily harvested for their thick, nacreous shells, which are used in the manufacture of buttons, ornaments, and other decorative products [4]. Intensive harvesting throughout its range has resulted in a significant decline in population densities and production yields [5]. In Pacific Island countries, annual production was estimated at approximately 1400 tons in the late 2010s, representing a 39% decrease from the 2300 tons estimate reported in 1996 [4]. This trend aligns with global declines exceeding 50% over the past two decades, primarily attributed to overexploitation [6].

Chuuk Atoll, located in the western region of the Federated States of Micronesia (FSM), is one of the largest and most ecologically significant atoll systems in the western Pacific. It comprises approximately 190 islands formed by volcanic activity during the Oligocene to Miocene epochs, arranged in an elliptical barrier reef that encloses a central lagoon [7]. The lagoon supports a diverse array of coral reefs, seagrass beds, and intertidal flats, which sustain a wide range of marine organisms and constitute a unique and highly biodiverse ecosystem [8,9]. In Chuuk, studies on reproductive cycle studies of *R. nilotica* have previously been conducted [10], and local efforts to develop production techniques and restore natural populations are currently underway. Despite the ecological and economic importance of the species, fundamental data on its morphological and genetic characteristics in Chuuk remain limited. This knowledge gap presents challenges for accurate species identification, understanding of population structure, and designing of effective conservation and resource management strategies. Therefore, a more comprehensive understanding of the Chuuk population is essential to support science-based restoration and sustainable utilization of *R. nilotica* in the region.

The use of mitochondrial genome data in taxonomic and phylogenetic studies has significantly enhanced resolution compared to analyses based on partial gene approaches, providing more comprehensive insights into molecular evolution, phylogenetics, and population genetics [11]. This approach is now widely employed in species identification and phylogenetic analysis of mollusks [12,13,14]. Although the mitochondrial genome of *R. nilotica* has been previously reported in Quanfu Island, South China Sea [15], the available record in NCBI remains unverified. In this study, we newly sequenced and annotated the complete mitochondrial genome of *Rochia* specimens from Chuuk Atoll. Based on the annotated mitogenome, we conducted molecular species identification and phylogenetic analyses to clarify the taxonomic status of the Chuuk population. The resulting data are expected to serve as a valuable reference for future research on resource conservation, physiological traits, and ecological dynamics of *Rochia* species.

## 2. Materials and Methods

### 2.1. Study Area and Sampling

Specimens of *Rochia nilotica* were collected by SCUBA diving from the northwestern fringing reef of Weno Island, Chuuk Atoll, FSM (7.458° N, 151.902° E), at 5–7 m depth (Figure 1A). Both adult individuals and juveniles (approximately one year old; with shell height of approximately 2 cm [16]) were sampled for morphological and molecular analyses (Figure 1B,C). Muscle tissue samples were dissected and stored at −20 °C until further processing for molecular analysis. All sampling and research activities were conducted under official authorization from of the Department of Marine Resources, Chuuk State Government, including permits for biological specimen collection, scientific research, and export.

### 2.2. Morphological Analysis

A total of 10 adult specimens were used for morphological classification. Shell biometric parameters, including shell height and width, were measured with digital calipers. For radula extraction, the buccal mass was dissected and digested in 2 M KOH at 60 °C for 2 h to remove soft tissue. The residual tissue was further cleaned with hydrogen peroxide. The cleaned radulae were air-dried and sputter-coated with platinum using Q150TS Sputter Coater (Quorum Technologies, Laughton, UK) for 60 s at a current of 20 mA. Scanning electron microscopy (SEM) imaging was performed using a MIRA3 Field Emission-SEM (Tescan, Brno, Czech Republic). Voucher specimens have been deposited at the Honam National Institute of Biological Resource (accession numbers: HNIBRIV 18781-18790).

### 2.3. Molecular Analysis

#### 2.3.1. DNA Extraction and Sequencing

Total genomic DNA was extracted from the foot muscle tissue of a single specimen using a Qiagen DNeasy Blood & Tissue Kit (Qiagen, Hilden, Germany), following the manufacturer’s protocol. The complete mitochondrial genome was sequenced on Illumina NovaSeq 6000 platform with 2 × 151 bp paired-end reads (Illumina, San Diego, CA, USA) at JS Link Inc. (Seoul, Republic of Korea).

Raw reads were processed using Trim Galore! (v0.6.7) [17] to remove adapter sequences and low-quality bases (phred score ≥ 30), with reads of at least 120 bp retained for downstream analysis. The read quality was assessed using FastQC (v0.11.9) [18].

In addition, partial COX1 and 16S rRNA sequences were newly generated from five additional *R. nilotica* individuals collected in Chuuk Lagoon, and these sequences were incorporated into the genetic analyses; detailed methods are provided in the Appendix A.

#### 2.3.2. De Novo Assembly and Annotation

De novo assembly of mitochondrial genomes was performed using MitoZ (v3.3) [19] in conjunction with MEGAHIT (v1.2.9) [20]. The initially assembled linear mitochondrial genome was used as a seed in NOVOplasty (v4.3.5) [21] to obtain the complete circular mitogenome. COX1 gene fragments were extracted from the assembled mitogenomes and compared against the NCBI nucleotide database using BLAST (v2.16.0) for species identification. Genome annotation was performed using MITOS2 (v2.1.6) [22] to annotate protein-coding genes and RNAs. Circos (v0.69-8) [23] was employed to visualize genome features.

### 2.4. Phylogenetic Analysis

Phylogenetic analyses in this study were conducted using two independent datasets. The first analysis employed a concatenated alignment of partial mitochondrial COX1 and 16S rRNA gene sequences to determine the genetic divergence among *Rochia* species. The second analysis was based on a concatenated alignment of all 13 protein-coding genes (PCGs) from the complete mitochondrial genome, providing deeper phylogenetic resolution within the superfamily Trochoidea. Multiple sequence alignments were performed using MAFFT (ver. 7.520) [24]. Partitioning schemes and evolutionary models were selected using PartitionFinder2 (v2.1.1) [25] under the Akaike’s Information Criterion (AICc) and the ‘--raxml’ option. The best-fit models for PCGs included GTR + I + G (for ATP6, COX1, COX2, COX3, CYTB, ND1, ND2, ND3, ND4, ND4L, ND5, ND6, and 16S rRNA) and GTR + G (for ATP8). Maximum likelihood (ML) analyses were conducted with RAxML-ng (v1.2.0) [26] using 1000 bootstrap replicates. Bayesian inference (BI) was performed using MrBayes (v3.2.7) [27] using two independent MCMC runs of 1 × 10^6^ generations, sampling every 1000 generations, with 25% burn-in. The resulting phylogenetic trees were visualized in FigTree (v1.4.4). For both phylogenetic analyses, *Rapana venosa* (NC_011193.1), a member of Muricidae, was used as the outgroup to root the trees.

## 3. Results and Discussion

### 3.1. Morphological Characteristics of Rochia nilotica from Chuuk Atoll

Specimens of *R. nilotica* collected from Weno Island exhibited morphological traits fully consistent with previous descriptions of *R. nilotica*. The shell is characteristically thick, smooth, and conical, lacking prominent external ridges (Figure 2A,B). The basal region near the aperture is slightly expanded, resulting in a rounded, outwardly curved appearance at the aperture margin (Figure 2A–C). Shell coloration comprises distinctive purplish-red radial streaks set against a predominantly white background, overlaid by a thin brown periostracum. This color pattern extends laterally and ventrally, while internally, the shells possess a highly lustrous nacreous layer. Additionally, a deep and open umbilicus, a prominent diagnostic feature distinguishing *Rochia* from closely related genera, such as Tectus within the family Tegulidae, was clearly observed (Figure 2D,H). Although the uppermost whorls in adult specimens were often eroded or obscured by encrusting algae, juvenile individuals clearly displayed sharply defined peripheral structures (Figure 2G,H). In particular, gear-like processes along the periphery and fine spiral riblets on the upper surface, as described by Okutani (2000) [28], were prominently observed in juveniles but became obsolete with growth, resulting in a smoother shell surface in adults. Shell proportion, expressed as the ratio of shell height to shell width (SH/SW), represent another key taxonomic characteristics within the genus *Rochia*.

Measurements from 10 adult individuals yielded SH/SW ratio of 0.95–10.4, closely aligning with previously reported values for *R. nilotica* populations in Indonesia [3,29] (Table 1).

This comparatively moderate slope clearly distinguishes *R. nilotica* from congeners with steeper profiles, including *R. conus* (Gmelin, 1791) and *R. maxima* (F. C. L. Koch, 1844), as well as from other species not included in our phylogenetic analyses, such as *R. elata* (Lamarck, 1822), *R. hirasei* (Pilsbry, 1904), and *R. magnifica* (Poppe, 2004) [30,31]. Radular morphology observed in Chuuk specimens was rhipidoglossan, typical of vetigastropods (Figure 3A). Each radular row consists of a small central tooth, five lateral teeth on each side arranged symmetrically (Figure 3B), and approximately 30–40 slender marginal teeth (Figure 3C,D). Due to the current lack of comparative radular data across other *Rochia* species, the detailed radular description provided here serves as an essential morphological baseline. Although the Chuuk population of *Rochia* exhibited no apparent morphological differences from previously described *R. nilotica* [3,29], molecular analyses revealed significant genetic divergence. This incongruence between morphological uniformity and genetic divergence suggests the possibility of cryptic speciation. To resolve species-level taxonomy and evaluate regional patterns, comprehensive morphological and molecular comparisons across *Rochia* populations throughout the Indo-Pacific region will be necessary.

### 3.2. Mitochondrial Genome Characterization and Phylogeny

The complete mitochondrial genome of *R. nilotica* collected from Weno Island, Chuuk Atoll, was assembled using a de novo approach and annotated as a circular molecule (Figure 4A). The final genome is 17,664 bp in length with a GC content of 34.15%. A total of 37 genes were identified, including 13 PCGs, 2 rRNAs, and 24 tRNAs, with gene duplications observed in trnE and trnG. A summary of gene content and genome features is provided in Table 2. The mitochondrial genome of *R. nilotica* has been deposited in the NCBI’s GenBank database (under accession number: PV929813).

A phylogenetic tree was inferred using 13 mitochondrial PCGs from 15 taxa within the superfamily Trochoidea, along with two outgroup gastropod species (Appendix A; Figure 4B). The *R. nilotica* specimen from Chuuk clustered most closely with *Rochia virgata* (KY205709.1) with strong statistical support [posterior probability (PP) = 1; bootstrap value (BV) = 100]. Additionally, the gene order (synteny) of the mitochondrial genome between the two species was conserved. These two *Rochia* species formed a monophyletic clade with *Tectus pyramis*, all belonging to the family Tegulidae. However, within Tegulidae, the phylogenetic topology revealed two distinct clades, indicating paraphyly of the family. One clade included the genera *Tegula*, whereas the other grouped *Rochia* and *Tectus* together. Notably, the *Rochia*–*Tectus* clade was recovered as a sister to members of the family Turbinidae (PP = 1; BV = 92), suggesting a closer evolutionary relationship between these two groups than with other Tegulidae genera. The topology of the mitochondrial genome-based phylogenetic tree constructed in this study was congruent with that of the concatenated COX1 and 16S rRNA gene-based tree, both supporting the paraphyly of the family Tegulidae. This finding is consistent with previous studies challenging the monophyly of this family and advocating for its taxonomic reevaluation based on mitochondrial genomic evidence [32,33,34]. These findings suggest that the current phylogenetic framework of the superfamily Trochoidea, which includes the families Tegulidae and Turbinidae, may not accurately reflect their evolutionary relationships. Although mitochondrial genome analyses provide higher phylogenetic resolution compared to single-gene approaches, their application can be limited by the lack of available comparative genomic data across taxa. By characterizing the complete mitochondrial genome of *R. nilotica* from Chuuk, this study contributes to filling that gap and provides a valuable reference for future phylogenetic, taxonomic, and evolutionary studies within Trochoidea.

### 3.3. Species Identification Based on Partial COX1 and 16S rRNA Gene Sequences

Mitochondrial COX1 and 16S rRNA gene sequences were extracted from the complete mitochondrial genome of the Rochia specimen collected from Weno Island, Chuuk Atoll, and additional partial COX1 and 16S rRNA sequences obtained from five newly sequenced Chuuk individuals were also included to refine the genetic comparison. To determine its phylogenetic placement, a concatenated gene tree was constructed using representative sequences from 30 taxa within the superfamily Trochoidea, with *R. venosa* from Muricidae designated as outgroups. The resulting phylogenetic tree, reconstructed under both BI and ML frameworks, placed the Chuuk specimen within the family Tegulidae, clustering closely with species of the genus *Tectus* (Figure 5). Among the seven currently recognized species in the genus *Rochia* (WoRMS; https://www.marinespecies.org/, accessed on 1 May 2025), four had available sequences and were included in the analysis, forming a strongly supported monophyletic clade (PP = 1; BV = 100). Although the Chuuk specimen showed no apparent morphological differences from previously described *R. nilotica*, its genetic profile was clearly distinct. In the phylogenetic tree, the Chuuk lineage formed a separate branch from *R. nilotica* specimens from Mo’orea Island (French Polynesia) and New Caledonia, which were closely grouped. Pairwise genetic distances further support this divergence. For the COX1 gene, the *p*-distance between the Chuuk and New Caledonia populations was 6.92%, and between Chuuk and Mo’orea (French Polynesia), it was 7.10–8.21% (Appendix A). In contrast, the distance between the Mo’orea and New Caledonia populations was only 0.16–0.33% (Appendix A), indicating high genetic similarity. For the 16S rRNA gene, sequence data were unavailable for the Mo’orea population; however, the *p*-distance between Chuuk and New Caledonia was 3.15%, further supporting the genetic distinctiveness of the Chuuk population (Appendix A).

The observed genetic divergence is likely driven by the long-term geographic isolation of Chuuk Atoll. Island populations inhabiting geographically isolated atolls can exhibit distinct genetic variation due to restricted gene flow [35], which plays a pivotal role in both evolutionary processes and conservation strategies. Chuuk consists of a central lagoon surrounded by more than 60 fringing islands, and while the internal reef slopes reach depths of up to 70 m, the outer slopes drop steeply into oceanic waters exceeding 4000 m in depth [36]. This abrupt bathymetric gradient may act as a significant barrier to larval dispersal and gene flow. Furthermore, the planktonic larval duration of *R. nilotica* is extremely short, approximately 4 days [37,38], which could exacerbate the effects of isolation by limiting the potential for long-distance dispersal [2]. These geographic and biological constraints likely contributed to the establishment of a genetically distinct population in Chuuk, despite morphological similarities to *R. nilotica* populations elsewhere. The observed divergence highlights the potential for cryptic speciation and underscores the importance of localized genetic assessments in informing conservation and resource management strategies in isolated reef systems such as Chuuk.

Our integrated morphological and genomic analyses revealed that the *Rochia* population from Chuuk shares morphological characteristics with *R. nilotica*, but exhibits notable genetic divergence, suggesting the presence of a cryptic species. The complete mitochondrial genome generated in this study provides a valuable genetic resource that can facilitate more precise species identification and further clarify potential species boundaries within the genus *Rochia*.

## 4. Conclusions

Our integrative approach combining mitochondrial genome sequencing, partial COX1 and 16S rRNA analyses, and detailed morphological observations revealed that the *Rochia* population from Chuuk Atoll exhibits clear genetic divergence despite morphological congruence with *R. nilotica*. The complete mitochondrial genome and phylogenetic analyses consistently placed the Chuuk lineage within the genus *Rochia* but separate from *R. nilotica* populations elsewhere in the Pacific, suggesting the presence of a distinct genetic lineage shaped by long-term geographic and dispersal isolation. These results not only highlight the potential for cryptic speciation within the genus but also emphasize the importance of using genomic data alongside traditional taxonomy for accurate species identification in geographically isolated reef ecosystems. The data generated in this study provide a valuable reference for future taxonomic, ecological, and resource management studies and support the need for expanded genomic comparisons across the Indo-Pacific, particularly including populations from the Indian Ocean, the species’ type locality.

## Figures and Tables

**Figure 1 animals-15-03471-f001:**
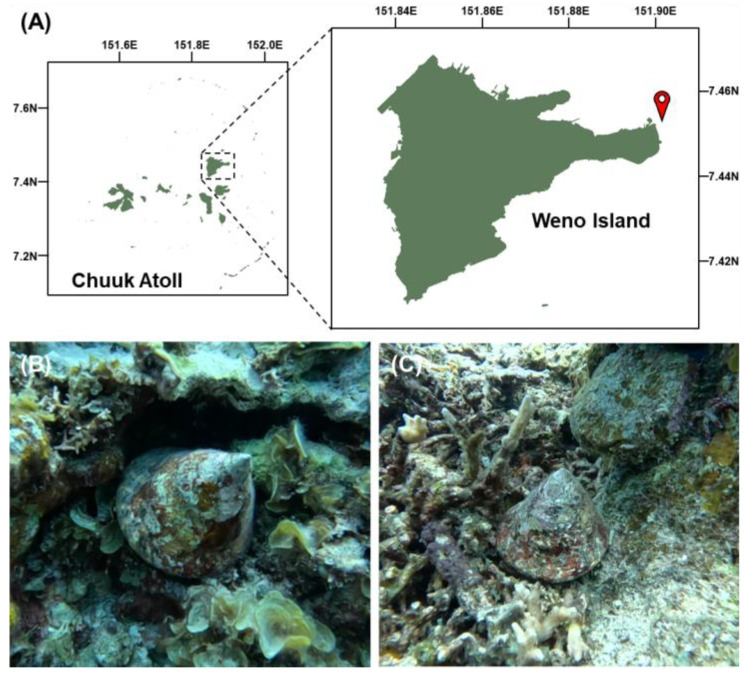
(**A**) Map of the study area showing Weno Island within Chuuk Atoll. The red marker indicates the sampling site. (**B**,**C**) Specimen of *Rochia nilotica* inhabiting coral reefs near Weno Island.

**Figure 2 animals-15-03471-f002:**
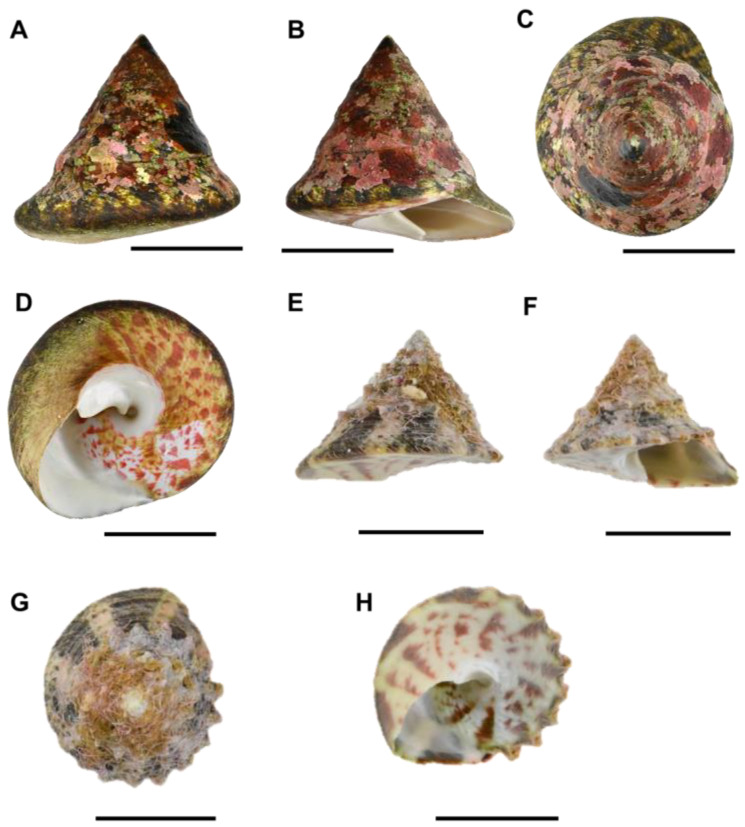
Images of *Rochia nilotica* shells. Adult: (**A**) dorsal view with aperture visible; (**B**) ventral view without aperture. (**C**) apical view; (**D**) abapical view. Juvenile (1-year-old): (**E**) dorsal view; (**F**) ventral view; (**G**) apical view without aperture; (**H**) abapical view with aperture visible. Scale bar: (**A**,**D**) 5 cm; (**E**–**H**) 2 cm.

**Figure 3 animals-15-03471-f003:**
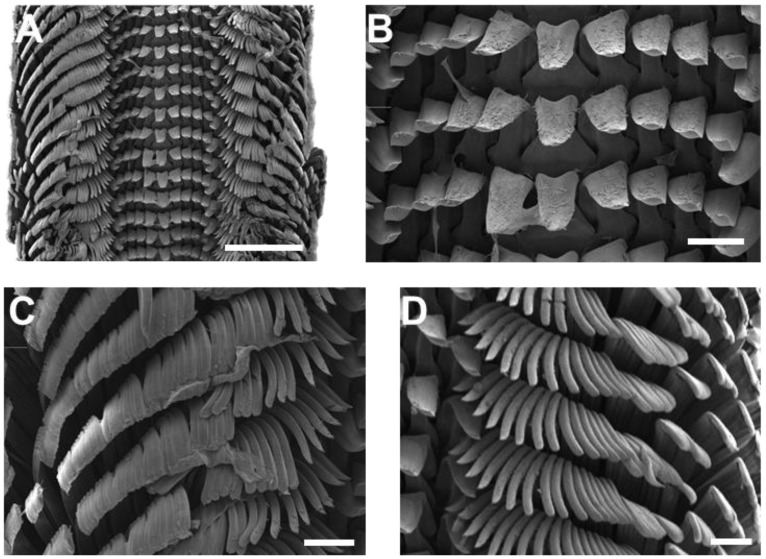
Radula morphology of *Rochia nilotica* from Chuuk Atoll. (**A**) Entire radula showing the rhipidoglossan arrangement. (**B**) Central region of the radula, including the central and lateral teeth. (**C**) Left marginal teeth in high magnification. (**D**) Right marginal teeth in high magnification. Scale bar: 200 μm.

**Figure 4 animals-15-03471-f004:**
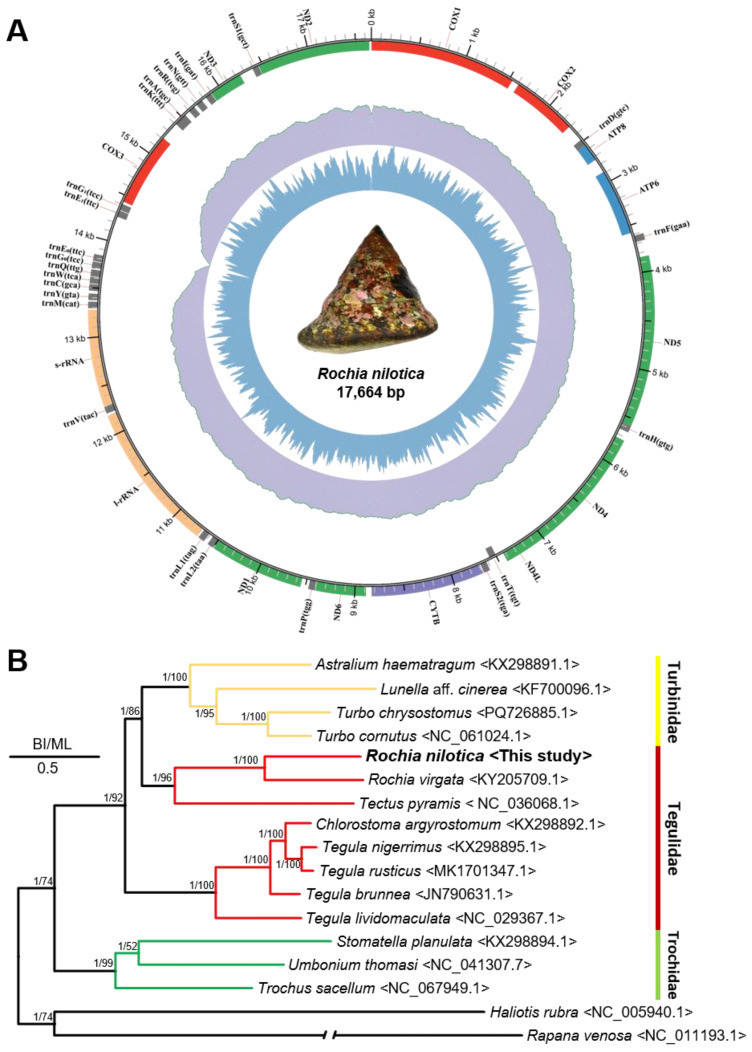
(**A**) Mitochondrial genome map of *Rochia nilotica* inhabiting Chuuk Atoll. The innermost blue bar plot represents GC content. The middle purple circle shows sequencing read depth. The outermost circle depicts gene order: red for COX genes, blue for ATP synthase, purple for cytochrome b (CYTB), and green for ND genes. The two rRNAs are in apricot, and the 22 tRNAs are in gray. (**B**) Phylogenetic tree constructed using 13 protein-coding genes (PCGs) from the mitochondrial genome. Vertical lines next to the species names indicate their family affiliations. The synteny of the 13 mitochondrial PCGs is aligned in a row to the right of the phylogenetic tree. Posterior probabilities (left) of Bayesian inference (BI) and bootstrap supports (right) for the maximum likelihood (ML) are indicated on the branches. The scale bar indicates 0.50 substitutions per site. Bootstrap supports <50% and posterior probabilities <80% are not shown.

**Figure 5 animals-15-03471-f005:**
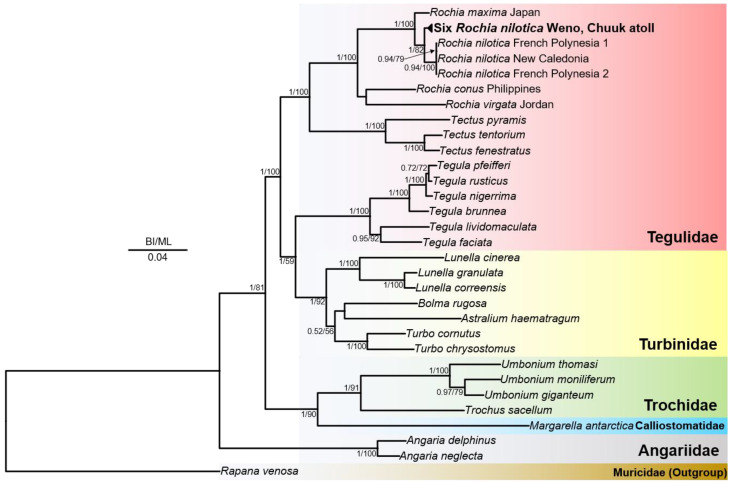
Phylogenetic trees of the superfamily Trochoidea based on concatenated COX1 and 16S rRNA sequences. Bayesian inference (BI) and maximum likelihood (ML) analyses were performed using the concatenated alignment of partial COX1 and 16S rRNA gene sequences. Specimens of *Rochia nilotica* from this study are indicated in bold. The best-fit substitution model (GTR + I + G) was applied, and node support values are shown as Bayesian posterior probabilities (0–1) and ML bootstrap values (0–100) based on 1000 replicates. GenBank accession numbers for all sequences used in the tree are listed in Appendix A.

**Table 1 animals-15-03471-t001:** Biometric information of *Rochia nilonica*.

Voucher Number	SH (mm)	SW (mm)	H/W
HNIBRIV 18781	101.39	104.83	0.97
HNIBRIV 18782	101.34	104.37	0.97
HNIBRIV 18783	91.73	100.85	0.91
HNIBRIV 18784	86.94	96.34	0.90
HNIBRIV 18785	92.58	96.27	0.96
HNIBRIV 18786	102.70	98.57	1.04
HNIBRIV 18787	90.70	91.99	0.99
HNIBRIV 18788	86.01	92	0.93
HNIBRIV 18789	88.49	94.53	0.94
HNIBRIV 18790	89.07	95.94	0.93
Average	93.10	97.57	0.95

SH, shell height; SW, shell width; H/W, shell height to width.

**Table 2 animals-15-03471-t002:** Summary of the mitochondrial genome of *Rochia nilotica*.

Gene Name	Location	Size	Start Codon	Stop Codon	Intergenic Region
Cytochrome c oxidase subunit 1	1–1536	1536	ATG	TAA	74
Cytochrome c oxidase subunit 2	1611–2297	687	ATG	TAA	179
tRNA-Asp	2477–2545	69	-	-	0
ATP synthase F0 subunit 8	2546–2734	189	ATG	TAA	149
ATP synthase F0 subunit 6	2884–3579	696	ATG	TAG	36
tRNA-Phe	3616–3683	68	-	-	159
NADH dehydrogenase subunit 5	3843–5576	1734	ATG	TAA	0
tRNA-His	5577–5643	67	-	-	90
NADH dehydrogenase subunit 4	5734–7131	1398	ATG	TAG	−7
NADH dehydrogenase subunit 4L	7125–7424	300	ATG	TAA	73
tRNA-Thr	7498–7568	71	-	-	48
tRNA-Ser	7617–7683	67	-	-	8
Cytochrome b	7692–8831	1140	ATG	TAA	55
NADH dehydrogenase subunit 6	8887–9393	507	ATG	TAA	4
tRNA-Pro	9398–9467	70	-	-	79
NADH dehydrogenase subunit 1	9547–10,494	948	ATA	TAA	4
tRNA-Leu	10,499–10,566	68	-	-	45
tRNA-Leu	10,612–10,679	68	-	-	15
16S ribosomal RNA	10,695–12,202	1508	-	-	22
tRNA-Val	12,225–12,294	70	-	-	3
12S ribosomal RNA	12,298–13,280	983	-	-	12
tRNA-Met	13,293–13,360	68	-	-	15
tRNA-Tyr	13,376–13,441	66	-	-	28
tRNA-Cys	13,470–13,535	66	-	-	6
tRNA-Trp	13,542–13,608	67	-	-	5
tRNA-Gln	13,614–13,682	69	-	-	11
tRNA-Gly	13,694–13,761	68	-	-	6
tRNA-Glu	13,768–13,837	70	-	-	426
tRNA-Glu	14,264–14,333	70	-	-	6
tRNA-Gly	14,340–14,407	68	-	-	24
cytochrome c oxidase subunit 3	14,432–15,211	780	ATG	TAG	195
tRNA-Lys	15,407–15,478	72	-	-	−5
tRNA-Ala	15,474–15,541	68	-	-	55
tRNA-Arg	15,597–15,665	69	-	-	34
tRNA-Asn	15,700–15,770	71	-	-	50
tRNA-Ile	15,821–15,889	69	-	-	4
NADH dehydrogenase subunit 3	15,894–16,247	354	ATG	TAG	142
tRNA-Ser	16,390–16,457	68	-	-	3
NADH dehydrogenase subunit 2	16,461–17,645	1185	ATG	TAG	18

## Data Availability

The mitochondrial sequence of *Rochia nilotica* is available from NCBI under accession no. PV929813.

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
