# Peer review of "Cryptic Divergence of Rochia nilotica (Gastropoda: Tegulidae) from Chuuk Lagoon, Federated States of Micronesia, Revealed by Morphological and Mitochondrial Genome Analyses"

_animals, 2025, doi:10.3390/ani15233471_

Round 1
Reviewer 1 Report
Comments and Suggestions for Authors
The paper titled “Cryptic Divergence of Rochia nilotica (Gastropoda: Tegulidae) in Chuuk Lagoon, Federated States of Micronesia, Revealed by Morphological and Mitochondrial Genome Analyses” described one mitochondrial genome of Rochia nilotica, which is high genetic divergence compared to the published COX and 16S RNA genes of Rochia nilotica. The phylogenetic relationship of Tegulidae was found non-monophyly. The paper is interesting to discuss the divergence of Rochia nilotica. However, I have some major comments in the paper.
- All species or genus names throughout the text should be italicized in accordance with standard taxonomic nomenclature conventions.
- Please combine the two paragraphs into one paragraph. eg. L90-L97, L117-128,
- Materials and Methods: Numerous sections lack clarity, and some descriptive content appears in the Results section. For example, was the phylogenetic tree constructed using the concatenated nucleotide sequences of the 13 protein-coding genes from the mitochondrial genome or those of cox1 and 16S rRNA? which species are used as outgroups?which tissue was used to draw DNA?
- Results and discussions:It is recommended to present the results and discussion as two separate sections.
- In Table 2: Using a single gene name is acceptable. Please refrain from using two gene names.
- In Figure 3: The published mitochondrial genome of Rochia nilotica could not be located in the available databases. It is recommended to include the R. nilotica species data from NCBI. If the mitogenome has not yet been annotated, it should be annotated before being incorporated into the analyses.
- In all figures, the figure are not clear; please replace those with a high-resolution image.
- I suggest conducting a molecular clock estimation of the divergence times among the cryptic species to provide evidence for whether the differentiation of species among the islands is caused by isolation on the same island. It is very important to discuss the cryptic species.
- Check the format of all references.
- Numerous additional revisions have been marked in the PDF document. Please review all annotations accordingly.

The English writing should be carefully reviewed, as it requires improvement.
Reviewer 2 Report
Comments and Suggestions for Authors
Please see attached file.

Round 2
Reviewer 1 Report
Comments and Suggestions for Authors
The paper has been well revised. However, minor revisions should be addressed in the References section.
1. In Reference 5, the full journal name should be used.
2. In References 19 and 21, "Nucleic acids research" should be corrected to "Nucleic Acids Research".
3. In Reference 23, "Genome research" should be updated to "Genome Research".
4. In References 24 and 25, "Molecular biology and evolution" should be revised to "Molecular Biology and Evolution".
5. In Reference 27, "Sys-tematic biology" should be corrected to "Systematic Biology".
6. In Reference 34, "Syst Biol" should be expanded to "Systematic Biology".
Reviewer 2 Report
Comments and Suggestions for Authors
Congratulations! The text now is ok for publication, but demanding the following minor modifications (which were indicated as corrected by authors, but actually not):
- Title: change "in Chuuk Lagoon" to "from Chuuk Lagoon".
- Fig. 4B: Trochidae, not Trochinidae.
- Lines 253-254: paraphyly, not polyphyly.
- Lines 282-284: 7.10-8.21%; 0.16-0.33%
- R. aff. nilotica still appears twice in the text after a search (CTRL+F)
I believe that's it. Thank you!
Maurício Fernandes (reviewer 2)
